# Contribution of Heparan Sulphate Binding in CCL21-Mediated Migration of Breast Cancer Cells

**DOI:** 10.3390/cancers13143462

**Published:** 2021-07-10

**Authors:** Irene del Molino del Barrio, Annette Meeson, Katie Cooke, Mohammed Imad Malki, Ben Barron-Millar, John A. Kirby, Simi Ali

**Affiliations:** 1Theme-Immunity and Inflammation, Faculty of Medical Sciences, Translational and Clinical Research Institute, Newcastle University, 3rd Floor Leech Building, Newcastle upon Tyne NE2 4HH, UK; irene.del_molino_del_barrio@kcl.ac.uk (I.d.M.d.B.); kcooke@lincoln.ac.uk (K.C.); Ben.Millar@newcastle.ac.uk (B.B.-M.); 2International Centre for Life, Bioscience Institute, University of Newcastle Upon Tyne, Newcastle upon Tyne NE1 3BZ, UK; annette.meeson@newcastle.ac.uk; 3College of Medicine, QU Health, Qatar University, Doha P.O. Box. 2713, Qatar; momalki@qu.edu.qa

**Keywords:** breast cancer, chemokines, metastasis, CCL21

## Abstract

**Simple Summary:**

Breast cancer is a leading cause of cancer-related deaths worldwide, predominantly caused by metastasis. Chemokine receptor CCR7 and its ligand CCL21 are implicated in the metastasis of breast cancer to the lymph nodes. Chemokine function is dependent upon binding to their specific chemokine receptors and negatively charged molecules on the cell surface (heparan sulphate). The role of heparan sulphate in CCR7-mediated lymph node metastasis was investigated by creating a non-heparan sulphate binding mutant chemokine CCL21. Mutant-CCL21 was tested in vitro in a range of assays, including cell migration, calcium flux and surface plasmon resonance spectroscopy. Mutant-CCL21 induced leukocyte chemotaxis in diffusion gradients but did not stimulate trans-endothelial migration of breast cancer cells. A murine model was used to assess the potential of mutant-CCL21 to prevent lymph node metastasis in vivo. Lymph node metastasis was significantly reduced by the administration of mutant-CCL21 compared to the control. Targeting chemokine–heparan sulphate interactions may be a promising approach to inhibit chemokine activity and metastasis.

**Abstract:**

Chemokine receptor CCR7 is implicated in the metastasis of breast cancer to the lymph nodes. Chemokine function is dependent upon their binding to both cell-surface heparan sulphate (HS) and to their specific receptors; thus, the role of HS in CCR7-mediated lymph node metastasis was investigated by creating a non-HS binding chemokine CCL21 (mut-CCL21). Mut-CCL21 (Δ103–134) induced leukocyte chemotaxis in diffusion gradients but did not stimulate trans-endothelial migration of PBMCs (*p* < 0.001) and 4T1-Luc cells (*p* < 0.01). Furthermore, the effect of heparin and HS on the chemotactic properties of wild-type (WT) and mut-CCL21 was examined. Interestingly, heparin and HS completely inhibit the chemotaxis mediated by WT-CCL21 at 250 and 500 µg/mL, whereas minimal effect was seen with mut-CCL21. This difference could potentially be attributed to reduced HS binding, as surface plasmon resonance spectroscopy showed that mut-CCL21 did not significantly bind HS compared to WT-CCL21. A murine model was used to assess the potential of mut-CCL21 to prevent lymph node metastasis in vivo. Mice were injected with 4T1-Luc cells in the mammary fat pad and treated daily for a week with 20 µg mut-CCL21. Mice were imaged weekly with IVIS and sacrificed on day 18. Luciferase expression was significantly reduced in lymph nodes from mice that had been treated with mut-CCL21 compared to the control (*p* = 0.0148), suggesting the potential to target chemokine binding to HS as a therapeutic option.

## 1. Introduction

Distant metastases, rather than the primary tumour, are the principal cause of death of most breast cancer patients. Chemokines and their receptors play essential roles in tumour biology, including leukocyte recruitment, tumour cell growth and survival, angiogenesis, and metastasis. The formation of site-specific metastases is largely dictated by chemokine receptors and their respective ligands. The dissemination of breast cancer cells to remote organs (bones, lungs, liver) is mostly governed by CXCR4 and its ligand CXCL12 [1,2]. However, lymph node invasion is usually the first step in the metastatic process, where CCR7-expressing tumour cells are attracted to its ligand CCL21 that is abundantly expressed in the lymph nodes [3,4,5]. This emulates the homing of naïve T-cells and dendritic cells when they cross the floor of the subcapsular sinus and enter the CCL21-enriched T cell zone in lymph nodes. 

Chemokine receptor CCR7 and its main ligands CCL19 and CCL21 are over-expressed in the lymph nodes. Of the two CCR7 ligands, CCL21 has more robust activities in dendritic cell migration [6] and is the most investigated in cancer. Indeed, the CCR7/CCL21 axis promotes the growth and metastasis of many tumour types, including melanomas [7], breast [2,5], thyroid [8], colon [9], head, and neck cancers [10]. CCR7 has emerged as an important marker in the prediction of axillary lymph node metastasis in breast carcinomas, particularly since CCR7 over-expression correlates with larger primary tumours, deeper lymphatic invasion and poorer survival rates [2].

In vivo studies revealed that metastatic tumour formation is decreased when CCL21 expression is knocked down in secondary lymphoid organs, since this diminishes both the chemotactic and antiapoptotic effects of CCR7-expressing tumour cells [11]. Similarly, the CCL21/CCR7 pair seems to play an important role in the lymphangiogenesis associated with pancreatic cancer [12]. The complete picture of the role and involvement of the CCL21/CCR7 pair in breast cancer is still undergoing development, but there are at least two areas in which this axis has been shown to be actively involved, including lymph nodes metastasis [13] and immune response modulation [14].

Most chemokines, including CCL21, form dynamic chemokine gradients, which are achieved by the binding of chemokines on glycosaminoglycans (GAGs) present on the surface of endothelial cells and in the extracellular matrix [15]. This creates an equilibrium of free and bound monomers and dimers, resulting in haptotactic and chemotactic gradients. This allows the directed movement of leukocytes from circulation to the site of injury via chemokine signalling through the G-protein coupled receptors (GPCR) [16]. Indeed, the elimination of all heparan sulphate from the lymph node endothelium abrogated CCL21 presentation and lymphocyte homing in a murine model [17]. GAGs are located primarily on the surface of endothelial cells, as macromolecular complexes with matrix proteins in the extracellular matrix (ECM). They can be divided into four groups: heparin/heparan sulphate, chondroitin sulphate/dermatan sulphate, keratan sulphate, and hyaluronic acid (a non-sulphated GAG, non-covalently attached to proteins) [18].

The role of HS in CCR7-mediated lymph node metastasis was investigated by creating a non-HS binding chemokine CCL21 (mut-CCL21)—this truncated CCL21 chemokine lacks residues 103–134 from the C-terminal end. A series of experiments was performed to determine how the deletion of the GAG-binding site altered the ability of CCL21 to stimulate chemotaxis within a concentration gradient generated by free solute diffusion. A further series of experiments was performed to compare the potential of WT and mut-CCL21 to stimulate the migration of cells across the endothelium. Finally, we used a murine model to assess the potential of mut-CCL21 in preventing lymph node metastasis.

## 2. Materials and Methods

### 2.1. Chemokine

The extended, basic C-terminus of CCL21 is reported to be necessary for GAG binding with deletions of CCL21 residues 103–134 resulting in greatly reduced or eliminated affinity for GAGs, as described by Hirose et al. [19]. Mut-CCL21, Δ103–134 (SDGGAQDCCLKYSQRKIPAKVVRSYRKQEPSLGCSIPAILFLPRKRSQAELCADPKELWVQQLMQHLDKTPSPQKPAQG) was synthesised (Almac, Penicuik, UK), purified by HPLC and the sequence verified by electrospray ionisation mass spectrometry; each of the reagents was endotoxin-free (<0.02 endotoxin units/mL). WT-CCL21 was sourced from R&D Systems.

### 2.2. Immunohistochemistry

Ethical approval for the use of paraffin-embedded samples from breast cancer tumours was approved under the REC reference 16/YH/0117 and samples isolated prior to the Human Tissue Authority (HTA), under the REC reference 06/Q0906/12. Informed consent was obtained from all subjects. Patient characteristics are described in Table 1.

Immunohistochemistry of 4 μM paraffin-embedded cancer sections was carried out using the ImmPRESS Polymer Detection Systems kit (Vector, Oxfordshire, UK) as per the manufacturer’s instructions without antigen retrieval. Anti-CCR7 (1:50, MAB197, R&D Systems, Abingdon, UK) antibody was incubated with the sections for 2 h at room temperature. The secondary antibody conjugated to the peroxidase was then added for 30 min followed by DAB (DAB Peroxidase (HRP) Substrate Kit, Vector) and counterstaining with haematoxylin.

### 2.3. Cell Culture

4T1-Luc cells were a generous gift from Prof. G. Lazennec (Centre national de la recherche scientifique (CNRS), Montpellier, France). Cells were cultured in DMEM’s media (Sigma-Aldrich, Dorset, UK) supplemented with 10% FBS, 5 mL of MEM Non-Essential Amino Acids Solution (100×), 100 U/mL penicillin, 100 μg/mL streptomycin, 0.146 g/L l-glutamine (Sigma-Aldrich) and G418 (500 µg/mL).

Cells were split the day prior to injection and harvested at around 40–50% confluency.

MDA-MB-231 and MCF-7 were cultured in Dulbecco’s Modified Eagle Media (DMEM) without phenol red (Sigma-Aldrich) supplemented with 10% FBS, 100 U/mL penicillin, 100 μg/mL streptomycin, and 0.146 g/L L-glutamine (Sigma-Aldrich). Primary human mammary epithelial cells (pHMEC) were cultured in HuMEC basal serum free media (Thermofisher Scientific, Paisley, UK).

### 2.4. Immunofluorescence

Cells were grown on 8-well chamber slides and fixed with ice-cold methanol. Primary antibodies for E-cadherin (Biolegend, 610181, 1:50 dilution), vimentin (Santa Cruz, Sc-7557-R, 1:100 dilution, Wembley, UK), ZO-1 (Abcam, Ab59720-50, 1:10 dilution, Cambridge, UK), mouse CCR7 (R&D Systems, MAB3477, 1:20 dilution) and human CCR7 (R&D Systems, MAB197, 1:20 dilution) were used with secondary Goat-anti mouse-FITC (Sigma-Aldrich, F0257, 1:100 dilution) and counterstained with DAPI, with no primary antibody as control. Slides were viewed using the Zeiss Axio Imager II microscope.

### 2.5. RT-qPCR

mRNA expression of CCR7 was assessed using the StepOnePlusTM PCR machine (Applied Biosystems, Waltham, MA, USA). Cells were lysed using the RNeasy Mini Kit (Qiagen, Hilden, Germany). A total of 1 µg RNA was used for reverse transcription with the Tetro cDNA synthesis kit (Bioline, London, UK), cDNA was used for qPCR with the SensiFast Probe Hi-ROX Mix (Bioline, London, UK) and CCR7 Taqman probes (Applied Biosystems, Waltham, MA, USA). RNA expression was normalised to GAPDH. Fold change expression was calculated in relation to human mammary epithelial cells (HuMEC) and quantified using the 2^ΔΔCt^ method.

### 2.6. Cell Surface Expression of Chemokine Receptors

The human CCR7-PE (FAB197P, R&D Systems) and mouse CCR7-PE antibody (FAB3477P, R&D Systems) used in this study were fully optimised before the collection of quantitative data. Briefly, 2 × 10^5^ cells were incubated with the corresponding antibody in 50 µL FACS buffer (2% BSA/PBS) for 30 min at 4 °C before washing and resuspending cells in 100 µL FACS buffer. Receptor expression was recorded using a FACS Canto II (BD Biosciences, Wokingham, UK) and analysed using FlowJo 7 (FlowJo, LLC, Ashland, OR, USA).

### 2.7. Surface Plasmon Resonance

Surface plasmon resonance was performed using a BIAcore 3000 as described previously [20,21]. Briefly, heparin or heparan sulphate was biotinylated ( (Merck, Gillingham, UK) at the reducing end and immobilised to a CM4 sensor chip. The chip surface was activated with 50 µL 0.2 M 1-ethyl-3-(3-dimethylaminopropyl)-carbodiimide (EDC) and 50 µL 0.05 M N-hydroxysuccinimide (NHS) before injection of 50 mL of streptavidin (0.2 mg/mL in 10 mM acetate buffer, pH 4.2). Remaining activated groups were blocked with 1M ethanolamine HCl, pH 8.5. Biotinylated heparan sulphate was immobilised by injecting 5 µL of 50 µg/mL HS in HBS with 0.3 M NaCl at a flow rate of 5 µL/min, and injections were repeated until a resonance unit (RU) increase of 200 was reached, after which the surface was then washed with 2 M NaCl. For binding assays, a range of chemokine concentrations (0–500 nM) were flowed across the chip at 25 µL/min for 5 min followed by a 500 s dissociation phase and a 2 min injection of 1M NaCl to regenerate the sensor chip surface. RU from a flow cell coated with streptavidin only was subtracted from the results from GAG coated flow cells and analysis was performed using BIAevaluation.

### 2.8. Calcium Signaling

PBMCs or MDA-MB-231 cells (10^6^/mL) were washed in HBSS supplemented with 1 mM CaCl_2_, 1mM MgCl_2_, 1% FBS (*v/v*) and incubated with 3 µm indo-1, AM for 30 min at 37 °C in the dark. Cells were then washed with supplemented HBSS at 400× *g* for 5 min, and resuspended at 3 million cells per 1.5 mL and left to rest for 30 min at 37 °C before analysis. Calcium flux was measured by FACS-Fortessa flow cytometry, using UV filter 530/30. Cells were studied for the effect of WT and mut-CCL21 on calcium flux. The basal concentration (HBSS, negative control) was subtracted to calculate the values. Finally, the ratio between the two wavelengths {355 450/50/355 530/30} was calculated and plotted against time in order to calculate the peaks of calcium release.

### 2.9. Chemotaxis Assays

Peripheral blood mononuclear cells (PBMCs) were isolated from whole blood taken from healthy volunteers. Ethical approval to obtain blood from healthy volunteers was granted by the Research Ethics Committee (12/NE/0121). PBMCs were isolated from heparinised blood using Lympholyte-H (Cedarlane Laboratories) as per manufacturer’s instructions.

Diffusion gradient chemotaxis assays were performed in a transwell system in 24-well companion plates (BD Falcon, Nottingham, UK) as described previously. Briefly, for PBMCs, 5 × 10^5^ cells were placed in the upper well of a 3 µm cell culture insert above (0–50 nM) WT or mut-CCL21 in 0.1% BSA/RPMI. The assay was incubated at 37 °C for 90 min. Migrated cells were quantified either by counting average number of cells per high power field or inserting them into FACS Tubes before CountBright™ Absolute Counting Beads were added. BD FACS Canto II Flow Cytometer was used to run the analysis. The number of migrated cells was calculated with the following formula:𝐿𝑎𝑏𝑒𝑙𝑙𝑒𝑑 𝑐𝑒𝑙𝑙𝑠 = (𝑁𝑜. 𝑜𝑓 𝑙𝑎𝑏𝑒𝑙𝑙𝑒𝑑 𝑐𝑒𝑙𝑙 𝑒𝑣𝑒𝑛𝑡𝑠 (𝑃1)/𝑁𝑜. 𝑜𝑓 𝑏𝑒𝑎𝑑 𝑒𝑣𝑒𝑛𝑡𝑠 (𝑃2)) 𝑁𝑜. 𝑜𝑓 𝑏𝑒𝑎𝑑𝑠 𝑎𝑑𝑑𝑒𝑑

For 4T1-Luc chemotaxis, fibronectin-coated 8 µm inserts were used. The undersides of filters were coated with 80 µL 2.5 µg/mL fibronectin (Sigma) for 30 min. Then, 2 × 10^5^ cells were placed in the upper well and allowed to migrate towards a range of chemokine concentrations for 6 h at 37 °C. Migrated cells were quantified by counting average number of cells/HPF.

In vitro, trans-endothelial chemotaxis assays were performed as above, except 72 h prior to the assay, 5 × 10^4^ HMEC-1 cells were cultured in 0.5 mL complete RPMI in cell culture inserts. All assays were performed in triplicate.

### 2.10. Creation of a Mouse Model of Breast Cancer

The experiments were carried out in full compliance of the Home Office Project License PPL 60/4497 within the Comparative Biology Centre (CBC) of Newcastle University. Mice were housed in a ventilated cage with a maximum occupancy of 6 mice and given food and water ad libitum.

We used a syngeneic transplantation model with murine cells. Female BALB/c mice (6–8 weeks) were injected with 50,000 4T1-Luc cells, and when the tumour became visible (day 7), one treatment group was injected i.v. in the tail vein with 20 µg mut-CCL21 in 100 µL PBS, whilst the second group was injected with 100 µL PBS alone. Treatment was carried out daily for 7 days, and tumour growth was monitored daily and measured using callipers. On day 18, all mice were humanely terminated as tumour size of one mouse surpassed the established guidelines, with tumour reaching 1 cm in two directions. Tumour volume was then calculated using the following formula:𝑇𝑢𝑚𝑜𝑢𝑟 𝑣𝑜𝑙𝑢𝑚𝑒 = 𝑙𝑒𝑛𝑔𝑡h × 𝑤𝑖𝑑𝑡ℎ × 𝑤𝑖𝑑𝑡ℎ/2

The liver, lungs, spleen, fat pad and axillary lymph nodes were harvested and prepared for either paraffin embedding, snap freezing or placed in 100 µL of RNAlater. Tissue was homogenised with the Qiagen Tissue lyser II and RNA extracted using RNAeasy kit.

### 2.11. Tumour Visualisation Using IVIS

In order to monitor tumour growth, mice were imaged weekly using the Xenogen IVIS Spectrum In Vivo Imaging System. Mice were weighted and injected i.p. with 150 mg/kg of 15 mg/mL luciferin before being anesthetised with 2.5% isoflurane and assessing their luminescence at 10 min using the IVIS. Luminescence was quantified by defining a region of interest (or ROI) and determining its average radiance using the Living Image^®^ software (Perkin Elmer, Cheshire, UK). The ROI was then applied to all the animals to compare the signals.

### 2.12. Statistical Analysis

All results are expressed as means ± SEM of replicate samples. The significance of changes was assessed by the application of an ANOVA with Bonferroni post-test or two-tailed Student’s t-test. All data were analysed using Prism 5.0 software (GraphPad software, San Diego, CA, USA). Luciferase expression values of axillary and branchial lymph nodes or all lymph nodes were averaged per mouse and statistical significance was calculated using a Kruskal–Wallis test.

## 3. Results

### 3.1. Expression of CCR7 in Breast Cancer

The expression of chemokine receptor CCR7 was assessed in patient samples with invasive ductal and lobular carcinoma with and without lymph node involvement (Figure 1 and Appendix A) using immunohistochemistry. The expression of the receptor was high, with variable staining patterns between cancer types. CCR7 staining in ductal carcinoma was mainly present in the cytoplasm and cell surface, particularly around the ducts, whilst in lobular carcinoma staining was mostly cytoplasmic but was also present in the nucleus of most infiltrating cells. In samples from patients with lymph node involvement, few CCR7 positive cells with cytoplasmic staining were observed.

### 3.2. Expression of CCR7 in Breast Cancer Cell Lines

The expression of CCR7 is shown to be upregulated in breast cancer tissue and to mediate metastasis to lymph nodes, but expression in breast cancer lines is poorly defined. Thus, CCR7 expression was assessed in a range of cell lines both at protein and mRNA level. Of the various cell lines tested, only 4T1 expressed moderate CCR7 in both flow cytometry and qPCR assays (Figure 2a–c).

Although 4T1 cells are epithelial in origin, many cancers undergo epithelial–mesenchymal transition and therefore express mesenchymal markers. These cells were further characterised for expression of epithelial markers. E-cadherin and ZO-1 were expressed at high levels at cell junctions. These cells also expressed mesenchymal marker vimentin, which extends in the form of filaments in the cytoplasm. The expression of murine CCR7 was also confirmed; however, its expression was not restricted to the cell surface but was also present in the cytoplasm, indicating that a proportion of the receptor is internalised. The specificity of staining was verified using human CCR7 antibody (Figure 2d).

### 3.3. Generation and Characterisation of Non-Glycosaminoglycan Binding CCL21

We synthesised human truncated-CCL21, lacking residues 103–134 from the C-terminal end to generate a mut-CCL21 (Figure 3a). Mut-CCL21 could bind its specific chemokine receptor but lacks residues that mediate its interaction with heparan sulphate.

To assess the impact of deletion on chemokine GAG binding, WT-CCL21 and mut-CCL2 binding was analysed by Surface Plasmon Resonance (SPR), in which reducing end biotinylated heparan sulphate (or heparin) was captured on top of a streptavidin sensor chip. This system mimics cell membrane-anchored proteoglycans, and SPR spectroscopy is used to measure changes in the refractive index caused by the interaction that occurs when WT or mut-CCL21 flows at a range of concentrations across the immobilised GAG sensor chips.

All chemokines interact with heparin, which serves as a model compound for heparan sulphate, the most ubiquitous class of GAG that is expressed on virtually every cell in the body. The resulting sensorgrams (Figure 3b) demonstrated that WT-CCL21 can interact with heparan sulphate, whereas no specific binding could be detected for mut-CCL21 at concentrations up to 50 nM. Heparan sulphate is the only GAG involved in CCL21 presentation in HEV in vivo and therefore most relevant physiologically. Similar results were observed for interactions with heparin (Figure 3b).

### 3.4. In Vitro Analysis of the Cellular Response to Mut-CCL21

***Ca2 + flux***: Indo-1-loaded cells were incubated with varying concentrations of either WT or mut-CCL21 and time-dependent changes in the concentration of intracellular-free calcium from baseline levels were monitored; the maximal value was recorded.

A representative fluorescence profile generated in response to the addition of 50 nM WT-CCL21 to labelled PBMC is shown in Figure 4a. No difference was apparent between the maximal response to WT-CCL21 and mut-CCL21 at 50 nM concentration (Figure 4b). Additionally, MDA-MB-231 cells with mut-CCL21 generated a smaller response than WT-CCL21 at 10 nM, whereas, there was no significant difference between the maximal response to wt CCL21 and the mut-CCL21 at 50nM concentration (Appendix A).

***Heparinoids***: To assess the functional competence of the impaired GAG binding, chemotactic migration was assessed in CCR7-expressing PBMCs. As seen in Figure 4c, 50 nM of WT or mut-CCL21 was incubated with increasing concentration of several GAGs (heparin, heparan sulphate, chondroitin sulphate A and chondroitin sulphate B) before assessing the migration of PBMCs in trans-filter assays. Heparin and heparan sulphate-bound WT-CCL21 in a dose-dependent manner achieved maximal inhibition at 250 and 500 µg/mL, respectively. However, a minimal effect was seen when competing with mut-CCL21, suggesting that the GAG-binding domain is key to abrogate chemotaxis using heparinoids. Furthermore, no effect was observed when competing with either chondroitin sulphate A or chondroitin sulphate B.

***Trans-filter chemotaxis***: To further assess the functional consequences of reduced GAG binding, chemotactic migration was assessed in CCR7-expressing human PBMCs, MDA-MB-231 cells and murine 4T1 cells. To assess this, PBMCs were allowed to migrate for 2 h through a 3 µm pore filter, whilst the latter two migrated overnight through an 8 µm pore filter towards a range of concentrations of WT or mut-CCL21.

Interestingly, both mut and WT-CCL21 induced PMBC migration at 10, 15 and 20 nM at similar levels (Figure 5a), and no difference was seen in MDA-MB-231 at 20 nM (Figure 5b). However, 4T1 cells’ migration was significantly reduced towards mut-CCL21 compared to the WT at both 30 nM and 50 nM (Figure 5c). As an example, the number of migrated cells observed with no chemokine, 30 nM mut-CCL21 and WT-CCL21 can be seen in Figure 5d.

***Trans-endothelial chemotaxis***: A further series of experiments was performed to compare the potential of WT and mut-CCL21 to stimulate the migration of PBMCs and 4T1-Luc cells across the endothelium. In these assays, HMEC-1 cells were cultured to confluency on the upper surface of each porous filter 72 h prior to the experiment. Similarly to the trans-filter chemotaxis, either WT or mut-CCL21 was then added to the basal compartment and PBMCs or 4T1-Luc cells were added above the apical surface of the model endothelium. In order to assess the trans-endothelial cell migration of 4T1-Luc cells through an HMEC-1 monolayer, 4T1-Luc cells were labelled with live dye orange CMRA and filters were visualised under a fluorescent microscope.

In contrast to the results for trans-filter migration for PBMCs, it was found that mut-CCL21 stimulated significantly reduced trans-endothelial cell migration of both 4T1 cells and PBMCs when used at 30 nM (Figure 6a–c), with levels remaining similar to the unstimulated cells. Similar results were also observed with MDA-MB-231 cells (Appendix A)

### 3.5. Effect of Mut-CCL21 in In Vivo Metastasis

To investigate the potential of mut-CCL21 to inhibit metastasis to the lymph nodes in vivo, a spontaneous metastasis, syngeneic murine model was established. Due to different 4T1 cell numbers used in the literature, we initially optimised the cell number to be injected in the mammary fat pad. Mice were injected with 50,000, 100,000, 500,000, 1 × 10^6^ and 2 × 10^6^ 4T1-Luc cells and imaged weekly with IVIS (Appendix A). Mice were terminated when tumours reached a size of 1 cm in both directions or 1.5 cm in one direction. Animals injected with 50,000 and 100,000 cells had a 100% survival up to day 17 (Appendix A), whereas animals with a higher number of cells had to be killed earlier due to tumour size. There is significant correlation between the tumour weight and the day of sacrifice (Appendix A).

Thus, mice were injected with 50,000 4T1-Luc cells, and once the tumour became visible on day 7, they were treated with mut-CCL21 or PBS (i.v.) daily for one week. Tumours were monitored non-invasively using IVIS and measured daily with callipers. Luminescence for all animals increased from day 1 to day 7. Interestingly, no statistical difference can be seen between day 7, 14 and 18 for any group (Appendix A). Furthermore, when tumours were harvested and weighed, no statistical difference was found between the two groups (Appendix A).

Animals were sacrificed at day 18 and lymph node positivity was assessed. To determine metastatic spread to the draining lymph nodes, RNA was extracted from the axillary and branchial lymph node (Figure 6d) and luciferase expression was quantified per mouse. Figure 6e shows the significant decrease in luciferase expression in all of the lymph nodes grouped together in mice treated with mut-CCL21.

## 4. Discussion

Chemokine receptor CCR7 has been linked to breast cancer metastasis in several studies, with the receptor being present in both normal and malignant mammary cells but markedly upregulated in the latter. Since then, CCR7 has been suggested as a biomarker for lymph node metastasis [2,22], but, unlike CXCR4, few potential therapies have been developed [23,24]. In vitro, most chemokines bind to GAGs such as heparan sulphate [25,26]; indeed, it has been seen that the presentation of chemokines by endothelial GAGs is an important requirement for the extravasation of leukocytes in vivo [27]. The hypothesis tested in the current study was based on the premise that cancer cells use the same mechanism as leukocytes to extravasate blood and lymph vessels, using CCL21′s presentation by GAGs to metastasise to the lymph nodes [17,28]. In order to compete with WT-CCL21, a mutant form of CCL21 with a truncated GAG-binding domain was created.

We found that the expression of CCR7 in breast cancer was mostly nuclear in lobular carcinoma and cytoplasmic in ductal carcinoma, with cytoplasmic staining being more prevalent. Indeed, Cabioglu [29] also reported that the staining for CCR7 was mostly cytoplasmic, with some expression in the nucleus. However, it was also reported that lymph node positive tumours showed more cytoplasmic CCR7 and that nuclear staining was exclusive to lymph node negative tumours. CCR7 expression was also assessed in a range of breast cancer cell lines: MDA-MB-231, MCF-7, SKBR3, T47D and 4T1. Low levels of mRNA were detected in all the cell lines; however, only 4T1 expressed detectable levels of CCR7 at the protein level. This is consistent with other studies that report low CCR7 mRNA levels in T47D, MDA-MB-231 and MCF-7 [1], low but inducible levels in MDA-MB-231 and MCF-7 [30], and no mRNA expression in MDA-MB-231 [31]. Pan et al. [32] also described low expression in MCF-7 cells, but conversely to our studies and other literature, reported high expression in MDA-MB-231 cells. In our hands, MDA-MB-231 expressed functional CCR7 as demonstrated by calcium mobilisation after CCL21 stimulation, but expression is low, as shown in the PCR studies.

We further assessed the expression of the epithelial markers E-cadherin and zonula occludens (ZO-1) and the mesenchymal marker vimentin in addition to human and murine CCR7 in 4T1 cells. In common with other studies, we found a high expression of E-cadherin and vimentin. This was also reported earlier [2] in tumours from breast cancer patients, where over 50% of the tumours expressed cytoplasmic CCR7, a phenomenon also observed by Liu et al. [33] and Andre et al. [19]. This cytoplasmic expression suggests receptor internalisation after ligand binding, and thus is a potential sign of receptor activation. Furthermore, CCR7 expression by 4T1 cells was also corroborated by Leung et al. [34] and Su et al. [35] at the protein level.

The effect of the truncation on CCL21 activity was assessed in MDA-MB231 cells, which express low levels of CCR7 as determined by PCR, 4T-1 cells and PBMCs. Calcium flux was not significantly different between mut-CCL21 and WT-CCL21 at 50 nM concentration in both the cell types. To confirm GAG involvement in CCL21-mediated chemotaxis, WT and mut-CC21 were incubated with heparin, heparan sulphate, and chondroitin sulphate A and B. WT-CCL21 showed dose-dependent inhibition of PBMC chemotaxis in response to heparin and heparan sulphate, whereas mut-CCL21 showed minimal effect. This suggests that the GAG-binding domain in CCL21 has a dominant role in abrogating chemotaxis in the presence of heparinoids. No effect was seen with either chondroitin sulphate A or B. This is in agreement with an earlier study which showed that CCL21 does not bind chondroitin sulphate A [36]. However, the effects of chondroitin sulphate B in chemotaxis and its binding to different chemokines are not fully explored. Overall, this shows that heparin and heparan sulphate have a higher binding affinity for CCL21 compared to chondroitin sulphate A and B.

Surface plasmon resonance (SPR) was carried out to assess the interaction between the ligand (GAG), which is bound to the sensor chip, and the analyte (CCL21), which flows over the surface. Mut-CCL21 did not bind to heparin, as opposed to the WT form. When heparin density was increased on the chip surface, a higher RU could be seen for both forms, but mut-CCL21 binding remained very low. The same binding pattern could be seen when heparan sulphate was immobilised instead of heparin, which is the only GAG involved in CCL21 presentation in HEV in vivo [37] and thus the most relevant physiologically. These results are in accordance with Hirose et al. [38], who also observed reduced binding to GAGs using an ELISA.

Compared to WT-CCL21, trans-filter chemotaxis of 4T1 cells towards mut-CCL21 was impaired, although cells still significantly migrated when compared to the no chemokine control. This lesser migration towards mut-CCL21 could be due to a less potent activation of the G-protein and lack of puncta formation, as discussed by Hjortø et al. [39]. However, conversely to our results, this study also showed that the chemotaxis of CCR7-expressing dendritic cells towards 10 nM mut-CCL21 was increased as compared to the WT, albeit this difference was lost at 100 nM. However, this study was carried out with dendritic and not cancer cells, and chemotaxis was assessed using an Ibidi 3D migration assay instead of a Boyden chamber, which are factors that could also explain the divergent results. Furthermore, other studies reported different results—on one hand, according to Hirose et al. [38], both chemokines elicit similar calcium flux levels, which should correspond to a similar chemotactic response [40]. On the other hand, Christopherson et al. [41] described that heparin can greatly inhibit trans-filter chemotaxis by binding to the GAG-region of CCL21, a similar strategy to our mut-CCL21. Indeed, previous studies in our group were also successful in inhibiting CXCL12-mediated trans-filter chemotaxis with heparinoids [42] or heparin [43].

Trans-endothelial migration studies show that migration towards mut-CCL21 was significantly lower than in response to WT-CCL21. Indeed, it has been reported that CCL21 presentation by GAGs is key for leukocyte rolling and diapedesis [17,28], and the same process is involved in lymph node extravasation during metastasis [44]. These results are supported by previous studies, which were also successful in inhibiting CXCL12-mediated [45] and CCL7-mediated [46] TEM using a similar mut-CXCL12 or mut-CCL7, which could not bind GAGs.

Further experiments were carried out to assess the potential of mut-CCL21 in a murine model. Mice were injected for 7 consecutive days from the moment the tumour was palpable, as that has been described as the moment when metastases start [47]. Given the very short half-life of chemokines in vivo [48] and that CCL21 does not desensitise CCR7, constant treatment would have been needed. However, truncated chemokines have been reported to have longer half-lives than their wild-type counterparts—for instance, WT-CXCL12 was cleared from the circulation 1 h post injection, whilst mut-CXCL12 could still be detected 24 h later [45]. With that in mind, and given that the tail vein injection number is limited before there is too much fibrosis, animals were administered with daily injections of 20 µg of mut-CCL21 to maintain a therapeutic level of mut-CCL21 in blood. Mice that had been treated with mut-CCL21 presented a significant decrease in the number of positive lymph nodes in comparison to the PBS control. Indeed, mice undergoing mut-CCL21 treatment were able to maintain or increase their weight during treatment, whilst the PBS group lost weight. A limitation of this study is that the treatment started when the tumour had been established, but no metastatic spread to the lymph nodes had been observed. Thus, mut-CCL21 was only assessed for its potential to prevent lymph node metastasis but not for its curative or prophylactic effect.

Although no side effects were observed in immunocompetent mice, it is important to note that CCL21 is crucial to the homing of naive T-cells to lymph nodes through the high endothelial venules [49]. Chronic disruption of this process could cause a reduced response towards inflammation, including the attack of the primary tumour itself. Whether alternative pathways can compensate for any reduced response towards CCL21 is still unknown, although it has been seen that mice lacking CCL21 still show a T cell immune response, albeit one that is delayed [50]. A possible strategy could be to encapsulate mut-CCL21 in sterically stabilised immunoliposomes, which are guided by antibodies that bind specifically to antigens on the tumour cell surface [51]. However, the target antigen must be chosen with care as chemokine internalisation is not desirable. Indeed, one of the main hurdles with nanoparticles for chemokine delivery is that they are designed to effectively penetrate the cell surface [52].

## 5. Conclusions

Overall, evidence from this study suggests that mut-CCL21 shows decreased binding to the GAGs present in the endothelial cells, and thus fails to create the chemotaxis gradient necessary for trans-endothelial migration and extravasation. However, mut-CCL21 can still bind and activate CCR7 as demonstrated by calcium flux, confirming that GAGs are not necessary for the binding and activation of the chemokine receptors in vitro. This impaired extravasation was also observed in vivo, where mut-CCL21 could reduce the number of metastatic lymph nodes in a breast cancer murine model. This highlights the therapeutic potential of disrupting GAG–chemokine interactions.

## Figures and Tables

**Figure 1 cancers-13-03462-f001:**
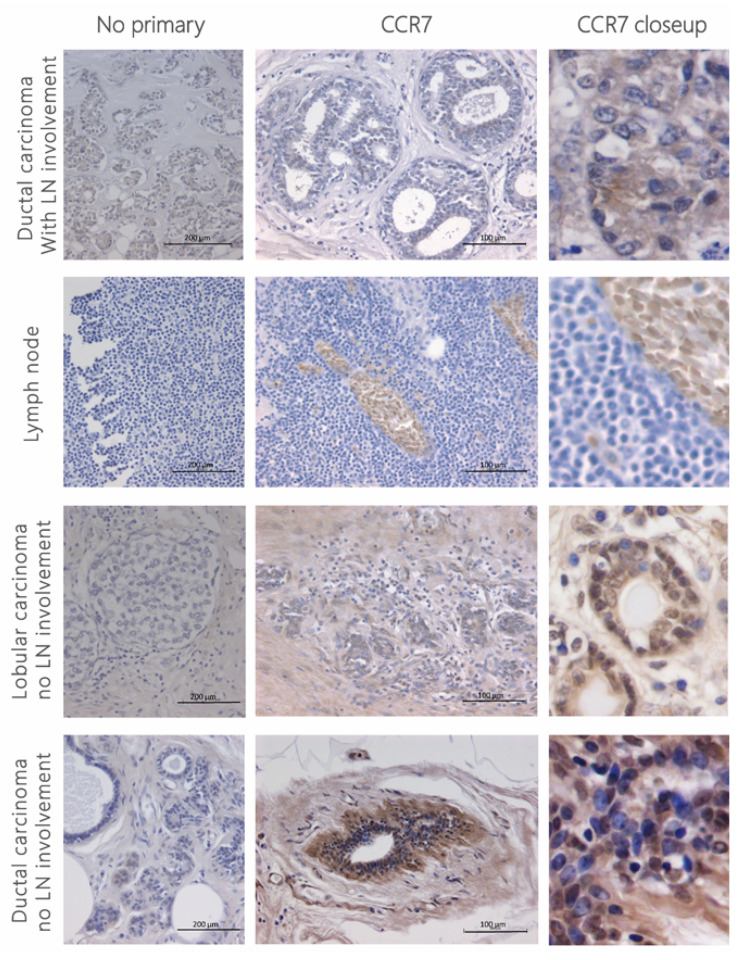
CCR7 staining in human breast cancer tissue. Four-micrometre sections from human invasive breast cancer were stained for anti-human CCR7 (1:50) using the ImmPRESS polymer detection kit following no pre-treatment for antigen retrieval. Signal was developed using DAB (brown stain) and counterstained with haematoxylin (blue). Microscopy images were taken at 20× magnification. No primary antibody was used as a control, depicted in the left panel at 10× magnification. Patient details can be found in Table 1.

**Figure 2 cancers-13-03462-f002:**
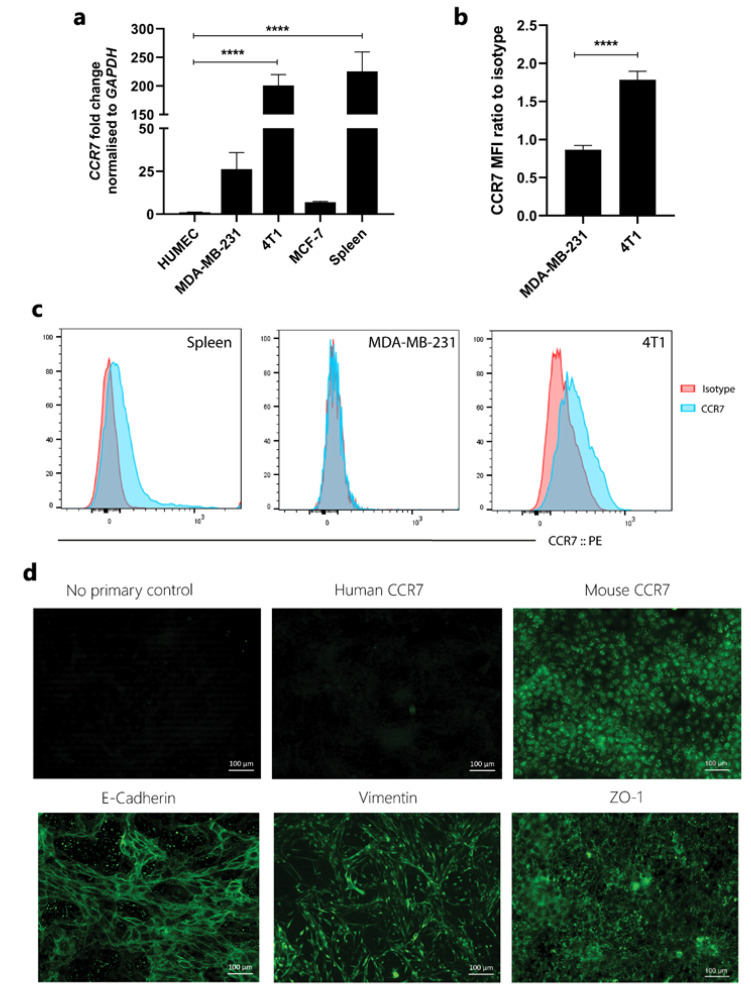
CCR7 expression in breast cancer cell lines. (**a**) CCR7 expression was assessed at RNA level in several breast cancer cell lines using RT-PCR and normalised to GAPDH. Fold change expression was calculated in relation to human mammary epithelial cells (HuMEC). Data represent the fold change ± SEM of three experiments performed in triplicate and statistical significance was assessed with one-way ANOVA with Tukey’s multiple comparison test. (**b**) CCR7 expression at protein level was assessed in MDA-MB-231 and 4T1 cells using anti-human or anti-mouse CCR7-Phycoerythrin (PE) antibody, respectively. Mean fluorescence intensity (MFI) of CCR7 was assessed by flow cytometry and normalised to the isotype control. Data represent the MFI ratio ± SEM of three independent experiments and statistical significance was assessed with Student’s *t*-test. (**c**) Confirmation of CCR7 antibody inter-species specificity, murine spleen, MDA-MB-231 and 4T1 cells stained with anti-mouse CCR7-PE antibody and assessed by flow cytometry. (**d**) CCR7 expression in 4T1 cells was further confirmed using immunofluorescence, and further characterisation was also carried out to ensure there was no phenotypic drift. Briefly, cells were grown in separate chambers, incubated with E-cadherin, vimentin, ZO-1 and human and mouse CCR7 antibodies, stained with FITC and counterstained with DAPI, with no primary antibody as control. Human CCR7 was included to assess the specificity of the antibody in a murine cell line. **** *p* < 0.0001.

**Figure 3 cancers-13-03462-f003:**
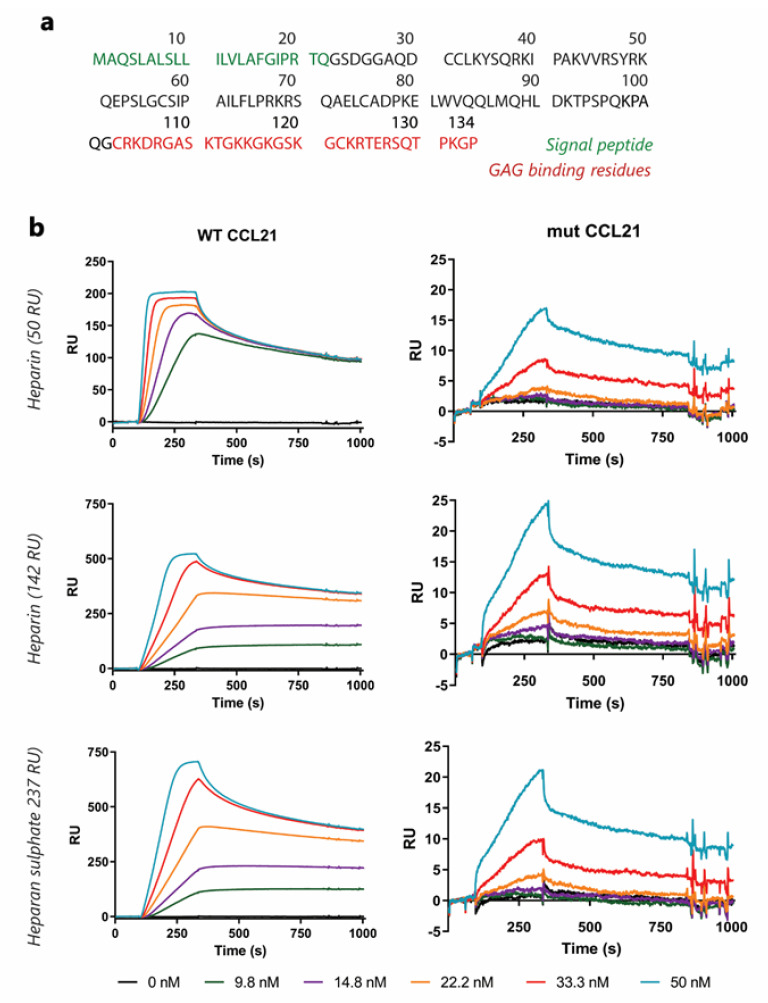
**3 Generation and characterisation of non-glycosaminoglycan binding CCL21** (**a**) CCL21 chemokine amino acid sequence. In order to create a non-HS binding CCL21, residues Δ103–134 (marked in red as the glycosaminoglycan binding domain) were eliminated from the C-terminal of CCL21 WT. The signal peptide (marked in green) will not be present in the mature protein. (**b**) Surface plasmon resonance (SPR) assay. Gold-coated Biacore chips were used to assess the alteration in GAG binding in mut-CCL21 as compared to the WT. (top) 0–50 nM chemokine flowed over immobilised heparin (50 RU bound) and the alteration in resonance units (RU) was recorded. (middle) 0–50 nM chemokine flowed over immobilised heparin (142 RU bound) and the alteration in resonance units (RU) was recorded. (bottom) 0–50 nM chemokine flowed over immobilised heparan sulphate (237RU bound) and the alteration in resonance units (RU) was recorded.

**Figure 4 cancers-13-03462-f004:**
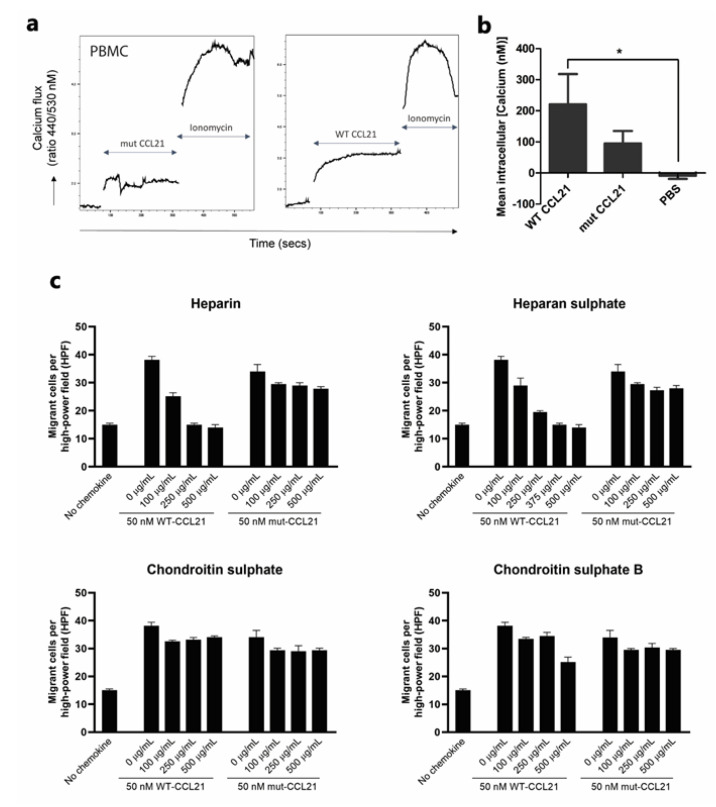
Biological activity of mutant and WT-CCL21 in PBMCs. Calcium flux. PBMCs were loaded with Indo-1am and stimulated with 50 nM of either PBS, WT or mut-CCL21, followed by ionomycin as a positive control. (**a**) Representative flow cytometry plots of the change in 450/50 to 530/30 ratio after stimulation with mut-CCL21 (left) or WT-CCL21 (right), followed by ionomycin. (**b**) Data represent the intracellular calcium concentration ± SEM of three independent experiments and statistical significance was assessed with one-way ANOVA with Tukey’s multiple comparison test. (**c**) Heparinoid inhibition. Inhibition of the chemotactic response of PBMCs towards 50 nM of either WT or mut-CCL21 after incubation with several concentrations of heparin (top left), heparan sulphate (top right), chondroitin sulphate A (bottom left) or chondroitin sulphate B (bottom right). * *p* < 0.05.

**Figure 5 cancers-13-03462-f005:**
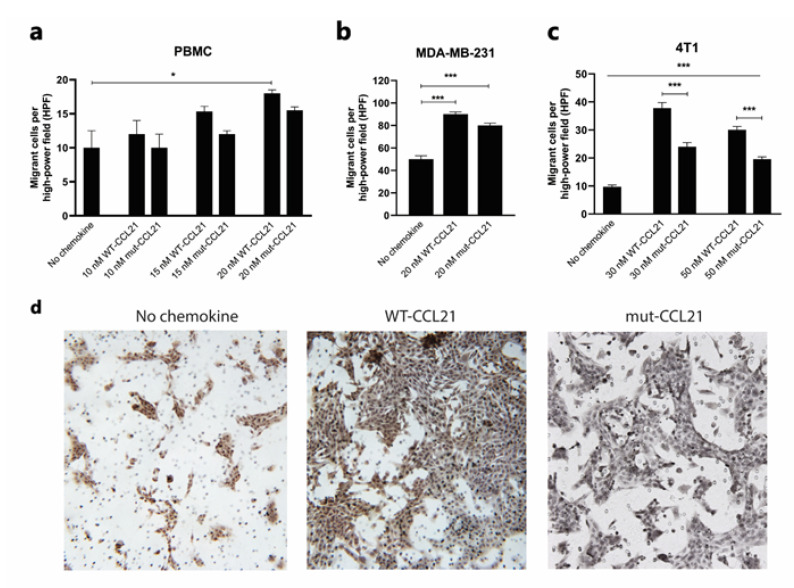
Trans-filter chemotaxis. Trans-filter chemotactic migration of PBMCs (**a**), MDA-MB-231 cells (**b**) and 4T1 cells (**c**) towards several concentrations of WT or mut-CCL21 after optimised incubation time. Graphs show the number of migrated cells adhered to the filter per high power field. Data represent the mean ± SEM of three independent experiments and statistical significance was calculated using one-way ANOVA with Tukey’s multiple comparison test. (**d**) Representative images of migrant 4T1 cells on the underside of chemotaxis filters. Migrant cells were counted using high power microscopy (×400) in four random fields on each filter. ***, *p* < 0.001; *, *p* < 0.05.

**Figure 6 cancers-13-03462-f006:**
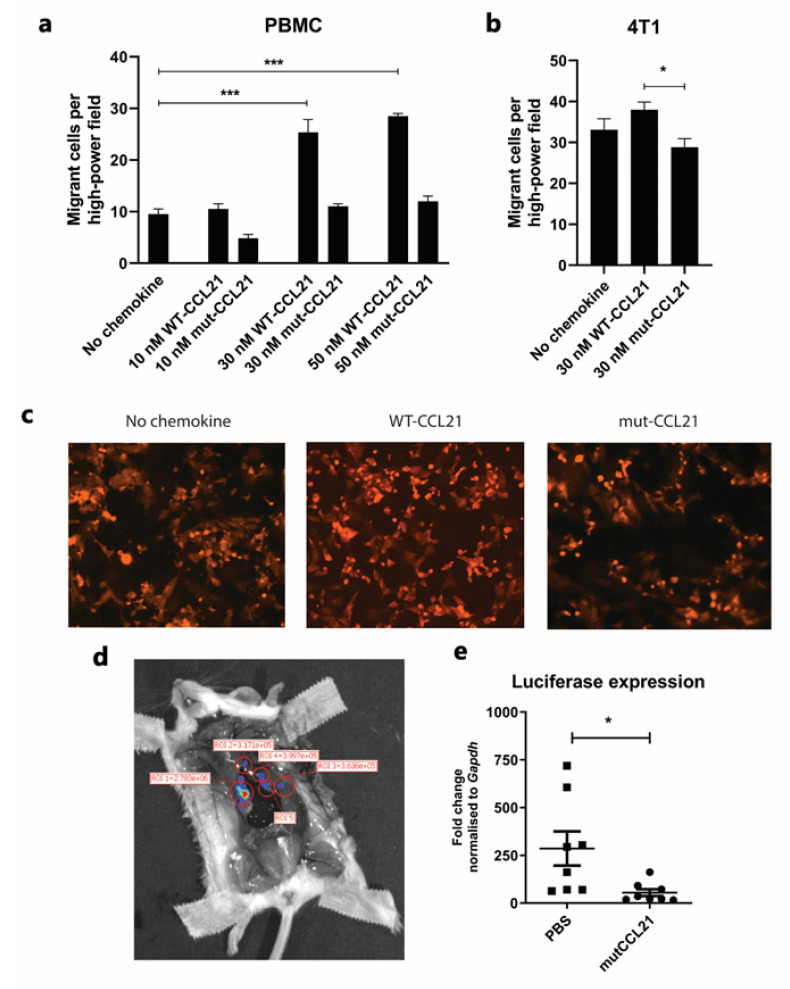
Trans-endothelial chemotaxis and in vivo migration. Trans-endothelial chemotactic migration of PBMCs (**a**) and 4T1 cells (**b**) though a HMEC-1 monolayer towards several concentrations of WT and mutated CCL21 after optimised incubation time. Graphs show the number of migrated cells adhered to the filter per power field. Data represent the mean ± SEM of three independent experiments and statistical significance was calculated using one-way ANOVA with Tukey’s multiple comparison test. (**c**) Representative images of migrant 4T1 cells on the underside of chemotaxis filters. Cells were labelled with CellTracker CMRA prior to migration to help distinguish them from the HMEC-1 monolayer. Migrant cells were counted using high power microscopy (×400) in four random fields on each filter. (**d**) IVIS visualisation of an open mouse. On day 18 post-4T1-Luc cell injection, luciferin was intraperitoneally injected and the mouse terminated 8 min later by overdose of anaesthetic. Skin was pulled away in order to reveal the internal organs and the tumour was surgically removed before visualising the “open mouse” 15 min post-injection. (**e**) RNA from the excised lymph nodes was extracted and average luciferase expression per mouse was calculated. Data represent the luciferase fold change normalised to GAPDH and statistical significance was calculated using a Kruskal–Wallis test (experimental group size: *n* = 8). ***, *p* < 0.001; *, *p* < 0.05.

**Table 1 cancers-13-03462-t001:** Patient details from paraffin-embedded tumour samples.

Figures	Patient	Location	Cancer Type	BR Grading	Lymph Node Involvement
**Figure 1**	Top	Breast tumour	Invasive ductal carcinoma (multifocal)	2	2 LN
	Top middle	Lymph node	Invasive ductal carcinoma	2	No
	Bottom middle	Breast tumour	Invasive lobular carcinoma	2	No
	Bottom	Breast tumour	Invasive ductal carcinoma	3	No
**Figure S1**	Sample 1	Breast tumour	Invasive ductal carcinoma	1	6 LN
	Sample 2	Breast tumour	Ductal carcinoma in situ	-	No
	Sample 3	Breast tumour	Invasive ductal carcinoma	3	No
	Sample 4	Breast tumour	Invasive lobular carcinoma	2	No
	Sample 5	Breast tumour	Invasive ductal carcinoma	2	No

## Data Availability

Data are contained within the article or Appendix A.

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
