# Peer review of "Contribution of Heparan Sulphate Binding in CCL21-Mediated Migration of Breast Cancer Cells"

_cancers, 2021, doi:10.3390/cancers13143462_

Round 1
Reviewer 1 Report
This paper by Molino del Barrio et al shows the contribution of heparin sulfate binding in chemokine CCL21-mediated migration of breast cancer cells to lymph node. To my knowledge it is the first time the contribution of the C-terminal part of CCL21 to breast cancer cells migration is shown. It is interesting because of the effect seems specific to breast cancer cell migration while not impacting leukocyte chemotaxis. This may open the way to future new selective treatments for breast cancer metastasis.
The manuscript is well presented and written. For me, 2 important references are missing : 1) it is a pity the recent review on CCL21/CCR7 axis in breast cancer progression made by the authors in Cancers is not cited. 2) Also the choice of the place of the truncation mutation (103-134) is not explained. This point should be discussed in light of the recent paper published in Biochemistry by Moussouras et al on the structural features of the CCL21 C-terminal tail.
Figure 2D : Immunofluorescence on 4T1 cells. The method is not described in the Method section, nor is the source of the antibodies used. It seems that the images were acquired on a wide field microscope. Confocal images especially for CCR7 labelling would be better because it is hard to see the intracellular labelling in the image presented.
Reviewer 2 Report
The manuscript by Molino del Barrio et al, try to describe the contribution of the heparan sulphate in CCL21-mediated migration of breast cancer. Although the manuscript could be of interest lack of solid experiments in human breast cell line. How the authors explain the results in figure 4? They observe a reduction of the cell migration upon treatment with higher concentration of H/HS; however the increase of GAG on endothelial cells blocks the rolling allowing the leukocytes extravasation and in this case should increase the migration, this point is not clear and maybe another model should be tested. Moreover, most of the experiments that show a link between the heparan-sulphate and CCL21 have been performed in murine or PBMC cells. In figure 5 the migration of MDA-MB-231 human breast cancer cells increases following treatment with CCL21 suggesting that although the levels of the CCR7 receptor are low at the basal point, with the stimulation the receptor is exposed on surface, but the mut-CCL21 has similar effect. It seems that in this model the there is no role of the heparan-sulphate in cell migration. Some experiments in human breast cancer are required.
Reviewer 3 Report
In this study, the authors investigated the contribution of heparan sulphate binding in CCL-21 to the metastasis of breast cancer. The expression of CCL-21 receptor, CCR7, were detected in human breast cancer samples and a murine breast cancer cell line, 4T1. The authors further synthesis a human truncated-CCL21 (mut-CCL21), which can bind to its specific chemokine receptor, but lack interactions with heparan sulphate. The chemotaxis of PBMC toward WT-CCL21 were inhibited by heparin and heparan sulphate, but their chemotaxis toward mut-CCL21 were not affect by these heparinoids. Additionally, the migration and trans-endothelial cell migration of 4T1 cells toward mut-CCL21 were reduced as compared to WT-CCL21. In the in vivo experiments, the authors found that treatment of mice with mut-CCL21 can reduce tumor metastasis to lymph nodes. Overall, the result of the manuscript provides a mechanism of how breast cancer cells metastasize to lymph nodes. Detail comments and suggestions regarding the manuscript are described below.
- Figure 2: The authors showed that the expression of CCR7 is low in human breast cancer cell lines (MDA-MB-231 and MCF-7). Can these cell lines migrate toward CCL-21?
- The in vivo experiment: 4T1 cells can cause spontaneous lung metastases after injection. Did the author exam lung metastasis in mice treated with PBS and mut-CCL21?
- The authors may need to show body weight of the mice after PBS and mut-CCL21 treatments.
- It is suggested that the authors describe how they perform cell culture (section 2.5) before creation of a mouse model of breast cancer (section 2.3) in the “Materials and Methods” section.
- The format of some references is not correct: Ref #17, #18, #19.
Round 2
Reviewer 2 Report
The authors’ responses are partially satisfactory.
The authors did not completely answer to the points 2 and 3. They did not presented an experiment such as the trans endothelial chemotaxis, in which the link between human breast cells, heparan sulphate and CCl21 is evident.
Reviewer 3 Report
The authors answered all my comments. No further questions.
Author Response
Reviewer 3 was satisfied with our earlier response, so no further comments to add.